# Synthesis of Carbonyl-Containing Oxindoles via Ni-Catalyzed Reductive Aryl-Acylation and Aryl-Esterification of Alkenes

**DOI:** 10.3390/molecules27185899

**Published:** 2022-09-11

**Authors:** Zhengtian Ding, Wangqing Kong

**Affiliations:** The Institute for Advanced Studies (IAS), Wuhan University, Wuhan 430072, China

**Keywords:** nickel catalysis, reductive coupling reaction, aryl-acylation, aryl-esterification, oxindoles

## Abstract

Carbonyl-containing oxindoles are ubiquitous core structures present in many biologically active natural products and pharmaceutical molecules. Nickel-catalyzed reductive aryl-acylation of alkenes using aryl anhydrides or alkanoyl chlorides as acyl sources is developed, providing 3,3-disubstituted oxindoles bearing ketone functionality at the 3-position. Moreover, nickel-catalyzed reductive aryl-esterification of alkenes using chloroformate as ester sources is further developed, affording 3,3-disubstituted oxindoles bearing ester functionality at the 3-position. This strategy has the advantages of good yields and high functional group compatibility.

## 1. Introduction

Carbonyl-containing oxindoles are ubiquitous core structures present in many natural products and pharmaceutical molecules, such as Convolutamydine A, Coixpirolactam A, AG-041R, Surugatoxin, and JMX0254, which show a wide range of biological activities (Figure 1) [1,2,3,4,5]. In addition, this framework is a very attractive synthon for the synthesis of other structurally complex indole alkaloids [6,7,8,9,10,11,12]. Consequently, it is highly desirable to develop efficient methods to access carbonyl-containing oxindoles from readily available chemical materials.

On the other hand, nickel-catalyzed reductive cross-coupling reactions pioneered by Weix [13] and Gong [14] et al., have received considerable attention over the past decade as they represent a powerful tool for the construction of diverse C–C bonds [15,16,17,18,19,20,21,22,23,24,25,26,27,28,29,30]. Compared with the classical redox-neutral protocol, this strategy allows reactions to proceed under mild conditions with high functional group tolerance, without the need for pre-preparation of sensitive organometallics. Furthermore, Ni-catalyzed reductive cyclization/cross-coupling reactions have also been developed, in which two C–C bonds are forged in one pot and the C(sp^3^) electrophilic fragment is generated in situ via intramolecular addition of a C(sp^2^) electrophile to an alkene. This method shows attractive application in the rapid construction of diverse functionalized heterocycles with sterically congested quaternary carbon stereocenters [31,32,33,34,35,36,37,38,39,40,41,42,43,44,45,46,47,48]. In 2019, our group reported a Ni-catalyzed reductive aryl-acylation of alkenes for the synthesis of carbonyl-containing oxindoles by using isobutyl chloroformate as carbonyl source (Figure 1A) [49]. However, this strategy is limited to the synthesis of dialkyl ketones. Subsequently, Wang et al., reported a nickel-catalyzed reductive aryl-acylation of alkenes by using ortho-pyridinyl esters as the acyl sources (Figure 1B) [50]. However, this method is restricted to the synthesis of aryl-alkyl ketones, and the use of acid anhydride as the acyl source failed to obtain the product. In order to overcome the shortcomings of the above methods, we hope to develop a general method to synthesize various carbonyl-containing oxindoles. Herein, we report Ni-catalyzed reductive aryl-acylation and aryl-esterification of alkenes, providing 3,3-disubstituted oxindoles bearing ketone and ester functionalities at the 3-position (Figure 1C).

## 2. Results

Our initial studies commenced with the cyclization/cross-coupling reaction of *N*-(2-bromophenyl)-*N*-methylmethacrylamide (**1a**) and benzoic anhydride (**2a**) utilizing NiBr_2_ as a catalyst, Mn as a reductant, and K_3_PO_4_ as a base in DMA at 80 °C. We expect that the reaction efficiency will be strongly ligand-dependent. As Table 1 shows, this turned out to be the case. After screening a variety of ligands (entries 1–8), we found that a rigid phenanthroline framework with electron-deficient carbonyl groups at the 4-positions (**L8**) was particularly suitable for our purpose, providing the desired ketone **3a** in 57% yield along with the reductive Heck product **4a** in 21% yield (entry 8). Different solvents were next investigated (entries 9–11), and MeCN was identified as the most effective solvent, affording **3a** in 67% isolated yield, while the reductive Heck product **4a** was reduced to 2% (Table 1, entry 10). The use of Zn^0^ instead of Mn^0^ resulted in little change in the yield of **3a**, but more side product **4a** was observed (compare entry 10 with 12). The reaction can be carried out at 60 °C without affecting the outcome of the reaction (entry 13). Finally, the best result was achieved using TBAB as an additive, providing **3a** in 85% yield with excellent chemoselectivity (entry 14). The reaction was carried out using 5 mol% nickel catalyst with only a slight decrease in product yield (entry 15). Finally, a series of control experiments confirmed that product was not formed in the absence of Ni^0^ catalyst and Mn^0^ (entries 16–17).

With the optimal conditions in hand, we turned our attention to validating the generality of the arylacylation protocol for the preparation of 3,3-disubstituted oxindoles with ketone functionalities at the 3-position (Figure 2). The substrate scope with respect to alkene-tethered aryl bromides **1** was first investigated. Different substitution patterns with electron-donating or electron-withdrawing groups on the aniline part were well tolerated, furnishing the corresponding oxindoles **3a**–**3i** in 51–90% yields. *N*-benzyl protected substrate was also accommodated, providing **3j** in 85% yield. The benzyl group can be easily removed to allow access to the N–H oxindole. The influence of the Cα substituents (R^3^) of the acrylamide double bond on the reaction outcome was examined. Methoxymethyl, benzyl, *n*-hexyl, and isopropyl all proceeded smoothly to give the corresponding oxindoles **3k**–**3n** in 60–77% yields. Remarkably, the pyridine backbone was also perfectly accommodated, furnishing aza-oxindole **3o** in 61% yield. In addition to aryl bromides, aryl triflates are also suitable electrophiles, as shown in the formation of **3p** and **3q**. We further investigated the scope of acid anhydrides. Both electron-deficient and electron-rich aryl anhydrides are well compatible with this reaction (**3r**–**3t**). Finally, phenylacetyl chloride was also found to be a suitable electrophile, providing the dialkyl ketone **3u** in 61% yield after slightly modifying the reaction conditions.

Encouraged by these results, we further hoped to achieve reductive aryl-esterification of alkenes. However, using the arylacylation reaction conditions in Figure 3, the corresponding ester product **6a** could not be obtained. A judicious screening of all the reaction parameters (see Appendix A) revealed that a combination of NiBr_2_ (10 mol%), 2,2′-bipyridine (20 mol%), Mn (3.0 equiv), and TBAB (0.5 equiv) in MeCN at 100 °C afforded **6a** in 75% isolated yield. With this reliable set of conditions in hand, we set out to explore the preparative scope of our catalytic aryl-esterification reaction. The aromatic ring of the aniline moiety with both electron-donating groups (Me, OMe) as well as electron-withdrawing groups (F, CF_3_) at the *para*-position was well tolerated to afford the corresponding oxindoles **6b**–**6e** in good yields. The *meta*- and *ortho*-substituted anilides generally react well to deliver the corresponding product **6f**–**6j** in good yields. Remarkably, pyridine backbone was also compatible to afford aza-oxindole **6k**, which has received particular attention due to its prominence in natural product and drug discovery programs. The cyclizative cross-coupling reaction of *N*-benzyl acetanilide with aryl chloroformate **5a** proceeded efficiently to provide **6l**. The influence of the Cα substituents (R^3^) of the acrylamide double bond on the reaction outcome was examined. Methoxymethyl, benzyl, *n*-hexyl, and isopropyl substituents were well compatible (**6m**–**6p**). Finally, the transformation is not limited to aryl chloroformates, and alkyl chloroformates can also react smoothly to obtain the corresponding alkyl esters (**6q**–**6s**).

## 3. Discussion

To gather direct evidence on the reaction intermediates involved in this transformation, we prepared σ-alkyl-Ni(II) complex **7** according to our previous report. The stoichiometric reaction of **7** with **5a** affords **6a** in 21% yield (Figure 4). The control experiment without nickel catalyst did not consume aryl bromide **1a** (Table 1, entry 16), indicating that the formation of aryl manganese species is unlikely. Taken together, we consider σ-alkyl-Ni(II)species **7** to be the key intermediate for this transformation.

On the basis of the experimental observations and previous studies [31,32,33,34,35,36,37,38,39,40,41,42,43,44,45,46,47,48,49,50], a plausible reaction mechanism is proposed (Figure 5). Oxidative addition of catalytically active nickel(0) **A** to aryl bromide **1** affords aryl-Ni(II) intermediate **B**, which undergoes intramolecular migratory insertion to give σ-alkyl-Ni(II) species **C**. Reduction of the intermediate **C** with Mn(0) affords σ-alkyl-Ni(I) intermediate **D**, which undergoes further oxidative addition to acid chloride **5** (or acid anhydride **2**) to form σ-alkyl-Ni(III)-carbonyl species **E**. Reductive elimination of intermediate **E** provides the final product and nickel(I) **F**, which regenerates the catalytically active nickel(0) upon Mn reduction.

## 4. Materials and Methods

### 4.1. General Procedure for the Synthesis of Ketones

An oven-dried sealed tube equipped with a PTFE-coated stir bar was charged with NiBr_2_ (10 mol%), 1,10-phenanthroline-5,6-dione (**L8**) (20 mol%), **1** (0.1 mmol, 1.0 equiv), manganese powder (3.0 equiv), TBAB (0.5 equiv), and K_3_PO_4_ (2.0 equiv). The sealed tube was evacuated and backfilled with argon (this process was repeated three times) and then MeCN (0.05 M) was added. This reaction mixture was stirred at room temperature for 15 min and then aryl anhydride **2** (2.0 equiv) was added. The reaction was heated at 60 °C for 36 h until the reaction was complete (monitored by TLC). The resulting mixture was purified by chromatography on silica gel, eluting with ethyl acetate/petroleum ether 1:20~1:5 (*v*/*v*) to afford the corresponding products **3**.

### 4.2. General Procedure for the Synthesis of Esters

An oven-dried sealed tube equipped with a PTFE-coated stir bar was charged with NiBr_2_ (10 mol%), bpy (**L1**) (20 mol%), acrylamide **1** (0.1 mmol, 1 equiv), manganese powder (3 equiv), and TBAB (0.5 equiv). The sealed tube was evacuated and backfilled with argon (this process was repeated three times) and then MeCN (0.025 M) was added. This reaction mixture was stirred at room temperature for 15 min and then acid chloride **5** (2~4 equiv) was added. Then, the reaction was heated at 100 °C for 36 h until the reaction was complete (monitored by TLC). The resulting mixture was purified by chromatography on silica gel, eluting with ethyl acetate/petroleum ether 1:20~1:5 (*v*/*v*) to afford the corresponding products **6**.

## 5. Conclusions

In summary, we have developed a nickel-catalyzed reductive arylacylation of alkenes using aryl anhydrides or alkanoyl chlorides as acyl sources, providing 3,3-disubstituted oxindoles bearing ketone functionality at the 3-position. Moreover, we further developed a nickel-catalyzed reductive arylesterification of alkenes using chloroformate as ester sources, affording 3,3-disubstituted oxindoles bearing ester functionality at the 3-position. This strategy has the advantages of good yields and high functional group compatibility. Future development of the asymmetric version is underway in our laboratory.

## Data Availability

The data presented in this study are available on request from the corresponding author.

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
