# Peer review of "Synthesis of Carbonyl-Containing Oxindoles via Ni-Catalyzed Reductive Aryl-Acylation and Aryl-Esterification of Alkenes"

_molecules, 2022, doi:10.3390/molecules27185899_

Round 1

Reviewer 1 Report

This manuscript describes synthesis of oxindoles contains keto/esters at 3-position via nickel-catalyzed reductive arylacylation/esterification of alkenes with the aid of acyl(anhydrides, alkanoyl chlorides)/ester sources(chloroformate). The defined studies could be an interesting extension of the current knowledge and novelty not extremely high. Since these are biologically valuable and pharmaceutical important scaffolds. However, in my opinion, the manuscript is suitable for publication in this journal after some concerns that the authors should be considered before its publication.

Comments

1.     In scheme, authors should mention relevant citations for each strategy and aryl acylation as well as esterification.

2.     Does authors check reactivity with nitro substituted systems??

3.     Authors used10-20 mol% is there any change if you use less mol% compared with utilized one.

4.     There are several typos in the manuscripts and space issues that need to be rechecked again.

5.     In SI file, authors have to include rf values, MPs, nature of the compounds such as crystalline/amorphous etc…, % of yields as well.

6.     Traces of impurities were found several places and unusual peaks in some of the cases that should be clearly mentioned in spectra’s whether solvent peaks?? or something else.

Author Response

Please check the response in the attachment.

Reviewer 2 Report

In this manuscript Kong and Ding reported the Ni-catalyzed reductive aryl-acylation and aryl-esterification reactions, providing oxindole derivatives. The optimization and the scope of the reaction was studied extensively. The mechanism of this reaction was also examined.

The study provides a new method for the synthesis of carbonylated oxindoles, which are frequently found in natural products. Since the study provides a new example in the active research area, I recommend this manuscript for publication provided that the following issues are addressed.

1) Page 2, Scheme 1B, "limited to alkyl ketone" should read "limited to aryl ketone"

2) Page 4, line 15, " benzoyl chloride " should read "phenylacetyl chloride"

3) SI, page S18. The C-F coupling should be reported for the 13C NMR spectra of 3t. Please re-analyze the spectroscopic data.

4) SI, page S67. The reported spectra of 6k are not satisfactory, and the 1H and 13C NMR spectra should be recorded with a sample with higher purity.

Author Response

(The authors gave the same response as above.)

Round 2

Reviewer 1 Report

The authors have made some of the requested modifications and i have no more comments.